# *In situ* or *Ex situ* Synthesis for Electrochemical Detection of Hydrogen Peroxide—An Evaluation of Co_2_SnO_4_/RGO Nanohybrids

**DOI:** 10.3390/mi14051059

**Published:** 2023-05-17

**Authors:** Constanza J. Venegas, Fabiana A. Gutierrez, Nik Reeves-McLaren, Gustavo A. Rivas, Domingo Ruiz-León, Soledad Bollo

**Affiliations:** 1Programa Institucional de Fomento a la Investigación, Desarrollo e Innovación, Universidad Tecnológica Metropolitana, Ignacio Valdivieso 2409, P.O. Box 8940577, San Joaquín 8320000, Santiago, Chile; cvenegasa@utem.cl; 2Laboratorio de Desarrollo Analítico y Quimiometría (LADAQ), Cátedra de Química Analítica I, Facultad de Bioquímica y Ciencias Biológicas, Universidad Nacional del Litoral, Ciudad Universitaria, Santa Fe 3000, Argentina; 3Consejo Nacional de Investigaciones Científicas y Técnicas (CONICET), Godoy Cruz CP C1425FQB, Buenos Aires 2290, Argentina; 4Department of Materials Science and Engineering, University of Sheffield, Sheffield S1 3JD, UK; 5Instituto de Investigaciones en Físico-Química de Córdoba (INFIQC), Departamento de Físicoquímica, Facultad de Ciencias Químicas, Universidad Nacional de Córdoba, Ciudad Universitaria, Córdoba 5000, Argentina; 6Laboratorio de Fisicoquímica y Electroquímica del Estado Sólido, Facultad de Química y Biología, Universidad de Santiago de Chile, Av. Libertador Bernardo O’Higgins n◦ 3363, Estación Central 9160000, Santiago, Chile; 7Centro de Investigación de Procesos Redox (CiPRex), Universidad de Chile, Sergio Livingstone Polhammer 1007, Independencia 8330015, Santiago, Chile; 8Advanced Center for Chronic Diseases (ACCDiS), Universidad de Chile, Sergio Livingstone Polhammer 1007, Independencia 8330015, Santiago, Chile

**Keywords:** nanohybrids, Co_2_SnO_4_, reduced graphene oxide, electrochemical detection

## Abstract

Nowadays, there is no doubt about the high electrocatalytic efficiency that is obtained when using hybrid materials between carbonaceous nanomaterials and transition metal oxides. However, the method to prepare them may involve differences in the observed analytical responses, making it necessary to evaluate them for each new material. The goal of this work was to obtain for the first time Co_2_SnO_4_ (CSO)/RGO nanohybrids via *in situ* and *ex situ* methods and to evaluate their performance in the amperometric detection of hydrogen peroxide. The electroanalytical response was evaluated in NaOH pH 12 solution using detection potentials of −0.400 V or 0.300 V for the reduction or oxidation of H_2_O_2_. The results show that for CSO there were no differences between the nanohybrids either by oxidation or by reduction, unlike what we previously observed with cobalt titanate hybrids, in which the *in situ* nanohybrid clearly had the best performance. On the other hand, no influence in the study of interferents and more stable signals were obtained when the reduction mode was used. In conclusion, for detecting hydrogen peroxide, any of the nanohybrids studied, i.e., *in situ* or *ex situ*, are suitable to be used, and more efficiency is obtained using the reduction mode.

## 1. Introduction

Co_2_SnO_4_ (CSO) is a ternary transition metal oxide with an AB_2_O_4_ inverse spinel structure. In this type of structure, B cations are found occupying tetrahedral and octahedral sites, while A cations occupy octahedral positions (B)_td_[AB]_Oh_O_4_; in normal spinel cations, A cations occupy tetrahedral positions and B cations occupy octahedral positions (A)_td_[B_2_]_Oh_O_4_ [1,2,3]. CSO stands out for its high thermal stability, low cost, and nontoxic nature. The presence of Co^2+^ in Co_2_SnO_4_ creates improved active sites for catalytic activity [4], which makes Co_2_SnO_4_ a good candidate for applications such as supercapacitors [5] and lithium batteries [6], but Co_2_SnO_4_ can be used for photooxidation of dyes [7] and water splitting processes [4], as well.

However, its application in sensors has been little studied. Wang et al. synthesized stabilized zirconia and micrometric CSO by a hydrothermal method and annealed it at 1000 °C for application as a H_2_S gas sensor [8]. K. Hwa et al. obtained CSO by a hydrothermal route followed by calcination, and under ultrasonication of CSO and reduced graphene oxide (RGO), a hybrid was obtained. This hybrid material was used for mesalamine voltametric detection [9]. M. Masjedi-Arani et al., through sonochemical synthesis in the presence of glucose, generated Co_2_SnO_4_ for subsequent mixing with RGO via a pregraphenization technique, followed by microwave synthesis, demonstrating the applicability of the hybrid material for electrochemical hydrogen storage applications [10]. In our previous work, we reported the substitution of cobalt in the ceramic phase Zn_2−x_Co_x_SnO_4_ (0.5 ≤ x ≤ 2.0) followed by an *ex situ* hybrid material with RGO. These phases were obtained by solid-state synthesis, presented particle sizes in the range of micrometers, and were tested for the amperometric detection of hydrogen peroxide. The hybrid with the highest proportion of Co^2+^ and Co_2_SnO_4_ had the best electrochemical response, proving to be an efficient electrocatalytic material when mixed with RGO [11]. On the other hand, CSO was also tested, forming a hybrid material with carbon nanotubes (CNTs) [12] and obtaining better analytical parameters than RGO. An explanation for these results may be that CNTs are capable of fully encapsulating the CSO particle due to their micrometric size, producing closer contact between the catalytic sites of both materials than that occurring between CTO and graphene sheets.

The main carbon-based materials used in the formation of nanohybrids are reduced graphene oxide [3,13], graphene oxide [14], carbon nanotubes [15], and carbon nitride [16], which have important roles in the development of sensors. These materials promote the homogeneity of the inorganic material in the hybrid material, increasing the area and improving the sensitivity and response times in the construction of an electrochemical sensor [17,18,19]. In addition, the electrocatalytic performance of the hybrid material was improved by taking advantage of the synergistic effect derived from the two constituent components [12,20]. However, this synergistic effect between both materials is determined by several factors, one of which is the synthesis method; these materials can be obtained by *in situ* or *ex situ* synthesis [3,21]. The *in situ* method involves the formation and growth of inorganic material in the presence of carbonaceous material, providing greater homogeneity. This method allows a wide variety of chemical and physical synthesis techniques that can be used, including methods of direct decomposition of precursors, solvothermal, hydrothermal, sol-gel, solid-state synthesis, and coprecipitation, among others [22]. In contrast, the *ex situ* method involves the mechanical mixing of both components of the hybrid material previously synthesized or commercially available, within the advantages it has is the possibility of using nanostructures with desired functionalities, and it is a good alternative for overcoming possible incompatibilities between the joint synthesis of nanomaterials [20].

In a previous study, Co_2_TiO_4_/RGO (CTO/RGO) nanohybrids were studied in our lab for electrochemical detection applications. CTO synthesis based on a one-step hydrothermal synthesis allowed us to obtain nanostructured CTO particles. The results showed that there is a synergistic effect between Co_2_TiO_4_ and RGO that is more significant when the hybrid is synthesized through an *in situ* methodology [3]. Considering the results of this study, the goal of this work was to synthesize CSO nanoparticles to obtain for the first time CSO/RGO nanohybrids via *in situ* and *ex situ* methods and to evaluate their performance in the amperometric detection of hydrogen peroxide.

## 2. Materials and Methods

### 2.1. Chemical Reagents

Tin chloride pentahydrate (Sigma Aldrich, St. Louis, MO, USA, 98%), cobalt chloride hexahydrate (Sigma Aldrich, St. Louis, MO, USA, 98%), sodium hydroxide (Merck, Darmstadt, Germany, Alemania, 98%), hydrogen peroxide (Merck, Darmstadt, Germany, Alemania, 30%), and reduced graphene oxide (RGO) from Graphenea^®^ and Nafion^®^ were purchased from Sigma Aldrich, St. Louis, MO, USA. A pH 12 NaOH solution was used as the supporting electrolyte. All solutions were prepared with ultrapure water (ρ = 18 MΩ cm) from a Millipore Milli-Q system.

### 2.2. Synthesis of the Nanohybrids

Co_2_SnO_4_ (CSO) synthesis: Nanomaterials were synthesized by a hydrothermal method. CSO was synthesized by modifying the synthesis reported by D. Yang et al. [23]. For this, 2 mmol of CoCl_2_∙6H_2_O and 1 mmol of SnCl_4_∙5H_2_O were used as sources of cobalt and tin, respectively, and they were dissolved in distilled water to form two transparent solutions. Under magnetic stirring, the two solutions were mixed, and a NaOH solution was added dropwise to obtain a concentration of 2.0 M. The solution was stirred for 2 h to later be transferred to a 23 mL stainless steel autoclave. The conditions determined for the synthesis of CSO were 24 h at 250 °C. After the reaction was finished, the autoclave was cooled to room temperature, and the resulting precipitate was centrifuged at 3000 rpm and washed with distilled water and ethanol repeatedly. Finally, it was dried at 80 °C in an oven for 12 h for subsequent characterization.

*Ex situ* synthesis (CSO+RGO): RGO was mechanically mixed with the CSO nanoparticles in an agate mortar at a ratio of 80:20.

*In situ* synthesis (CTO/RGO): The same procedure was used for the synthesis of CTO, but after mixing all the precursors, RGO was added. The hydrothermal syntheses then proceeded using the same time and temperature conditions.

### 2.3. Modification of Glassy Carbon Electrodes (GCEs) with Hybrid Nanomaterials

To modify the GCEs, 2.0 mg of hybrid material (CTO + RGO or CTO/RGO) was dispersed in 1.00 mL of Nafion^®^ (0.2% *v*/*v* in ethanol) by sonication for 30 min. Before surface modification, the GCEs were polished with 0.3 and 0.05 µm alumina slurries for 1 min. Modification of the hybrid nanomaterial dispersion was achieved by drop casting 10 µL of the dispersion onto the GCE, followed by evaporating the solvent at room temperature.

### 2.4. Characterization

X-ray diffraction (XRD) data were collected in a PANalytical X’Pert^3^ Powder diffractometer with Cu αK radiation. Data analysis was performed using the graphical user interface for GSAS experiment, EXPGUI [24] and GSAS (Generalized Structure Analysis System) [25]. Raman measurements were recorded with a WiTec Alpha 300 Raman-AFM using a 532 nm laser. The surface morphology was obtained by scanning electron microscopy (SEM) (TESCAN, Vega 3 model). Histogram size distribution was calculated using ImageJ software. A compositional study was conducted by energy dispersive spectroscopy (EDS) using a Bruker probe (model QUANTAX 400a series). Thermogravimetric analysis was carried out with a Shimadzu DTG-60 instrument in a flowing air atmosphere with an increase of 10 °C min^−1^ in platinum canisters. The reference used was alumina.

Textural properties were obtained from the adsorption–desorption isotherm of N_2_ at 77 K, which was carried out using a Micromeritics 3Flex. The sample was previously degassed for 10 h at 76.9 K under vacuum using a Micromeritic SmartVacPrep. The specific surface area was determined from the adsorption branch in the range of 0.05 ≤ p/p0 ≤ 0.25 using the Brunauer–Emmett Teller (BET) theory [26].

Electrochemical impedance spectroscopy (EIS), cyclic voltammetry, and amperometry were performed with a three-electrode cell. Ag/AgCl, 3.0 M KCl (CH Instrument, Bee Cave, TX, USA), and platinum wire were used as the reference electrode, electrolyte, and auxiliary electrode, respectively. The working electrode was a glassy carbon electrode (GCE, CH Instrument) modified with our hybrid materials. Electrochemical impedance spectroscopy (EIS) measurements were performed with an Autolab PGSTAT 128 N potentiostat (EcoChemie, Herisau, Switzerland) in a frequency range of 10,000 Hz–0.1 Hz (amplitude: ∼10 mV). The redox probe was 0.010 M hydrogen peroxide prepared in pH 12 NaOH, and the working potential was −0.400 V. The impedance spectra were analyzed using Z-view software. Amperometric and voltametric experiments were performed using a Palm Sens potentiostat (Houten, The Netherlands). The amperometric experiments were conducted in deoxygenated pH 12 NaOH solution at −0.400 and 0.300 V by applying the desired working potential and allowing the transient currents to decay to a steady-state value before the addition of 0.10 mM H_2_O_2_, with subsequent current monitoring.

## 3. Results and Discussion

### 3.1. Characterization of CSO/RGO and CSO+RGO Hybrid Nanomaterials

The phase formation of Co_2_SnO_4_ (CSO) was carried out by X-ray diffraction (XRD). Figure 1a shows the phase analysis for CSO+RGO and CSO/RGO. The observed 2θ values were indexed to the cubic spinel structure CSO, space group Fd3m, for database entry into the International Centre for Diffraction Data (ICDD) [ICDD #00-029-0514]. The diffraction planes are well defined, and no impurities or extra planes were detected. No diffraction plane (002) related to RGO (2θ = 24°) was detected in the hybrid materials, which may result because of the low content or amorphous form of graphene.

Similar cell parameters were determined by the Rietveld method.

a = 8.6435(1) Å and a = 8.6694(1) Å attributed to CSO present in the CSO+RGO and CSO/RGO nanohybrids, respectively. Applying the Scherrer equation on the three highest Bragg peaks, the crystallite average size was estimated (Table 1), concluding that both nanohybrids have similar and nanosized crystallites.

To confirm the percentage of RGO in the *in situ* nanohybrid, thermogravimetric analysis (TGA) was carried out. Appendix A shows the loss in mass as a function of temperature, where the strongest loss was observed at approximately 450 °C, which is attributed to the mass loss of carbonaceous materials [27]. This result allowed us to determine an approximate 20% of RGO; that is, there was no loss of carbonaceous material during the *in situ* synthesis process of the nanohybrid.

To verify the integrity of RGO in both hybrids, Raman spectroscopy was used. Figure 1b shows the Raman spectra for both nanohybrids in the Raman shift between 1000 and 3500 cm^−1^. A vibrational mode at 1350 cm^−1^ corresponding to the D band, related to vacancy defects and the degree of structural disorder that occurs in structures with C-C bonds with sp^3^ hybridization, was observed. Additionally, a second characteristic vibrational mode at 1580 cm^−1^ corresponding to the G band associated with the sp2 vibrational mode in the plane was present. Finally, as a sign of the crystallinity of the material, a third G′ mode at 2690 cm^−1^ of lower intensity was observed, associated with a phononic vibration that is typical of materials that present sp2 hybridization [28,29,30]. The integrated intensity ratio, I_D_/I_G_, of the D and G bands widely used to characterize the degree of defects in graphitic materials [30,31] was 1.79 and 1.77 for CSO+RGO and CSO/RGO, respectively. These values were similar to those obtained for RGO alone (1.80). In conclusion, the graphitic structure remains intact and is in concordance with the reported values, which fluctuate from I_D_/I_G_ 1.57 to 2.20 [28]. Thus, our materials have a graphitic structure with a high degree of defects, confirming the presence of RGO in both nanohybrids.

On the other hand, group theory analysis predicts 42 vibrational modes in spinel, but only five of them are active in Raman spectroscopy, Γ = A1g + Eg + F2g [32,33,34]. Figure 1c shows a four-phonon mode spinel structure characteristic of CSO in the CSO/RGO and CSO+RGO nanohybrids, 630 cm^−1^, 510 cm^−1^, 310.7 cm^−1^ and 216.7 cm^−1^ for A1g, F2g(2), Eg and F2g(1), respectively. The peaks are not very well defined, attributed to the dark color of cobalt oxide, which reduces the penetration depth of the excitation laser beam, which also makes it difficult for not all the vibrational modes to be observed, in this case, vibrational mode F2g(3).

According to the characterization by Raman spectroscopy, it can be concluded that there is no bond formation between CSO and RGO because no additional vibrational modes are observed.

The specific surface area of the hybrid nanomaterials was studied by the Brunauer-Emmett-Teller (BET) method. The nitrogen adsorption isotherm of CTO was compared to *in situ* and *ex situ* hybrids (Figure 1d). The type II isotherm for CSO corresponds to nonporous or macroporous materials with a surface area of 10 m^2^g^−1^ and a total pore volume of 0.080 cm^3^/g. In contrast, both nanohybrids present type IV BET isotherms, corresponding to porous materials. Textural parameters such as surface area, pore size and total volume size are shown in Table 1, with no significant differences between the nanohybrids. Clearly, the incorporation of RGO favors the formation of micropores and mesopores, increasing the total volume of pores and the BET surface by at least two times.

Nanohybrid morphology, studied by SEM microscopy, revealed a cubic type of structure corresponding to the ceramic material (Figure 2a,b), without observing areas that indicate the presence of RGO; therefore, we conclude that CSO is on sheets of carbonaceous nanomaterial. From a set of SEM-STEM micrographs, the particle size for CSO in each nanohybrid was determined. The distribution histograms are shown in Appendix A, confirming the nanostructured size of the spinel. Sizes of 83 ± 30 nm and 67 ± 28 nm for CSO in CSO+RGO and CSO/RGO, respectively, were determined. The wide range of size dispersion is attributed to the hydrothermal synthesis route used [35].

Surface mapping of Co, Sn and O was carried out to determine the distribution of CSO on the hybrid nanomaterials. SEM-EDX micrographs of each of the chemical elements are shown in Figure 2b for CSO+RGO and Figure 2c for CSO/RGO. A clear more homogeneous distribution of CSO on the material was observed in the case of the *in situ* nanohybrid than in the case of the *ex situ* nanohybrid. Since the formation of the nanohybrid material via *in situ* synthesis implies the formation and growth of the cobalt spinel in the presence of the RGO, its presence allows a uniform distribution of the polycrystalline nanomaterial to be obtained, as was described previously in other works [3,10,22]. Additionally, for comparative purposes, EDX spectra for CSO/RGO and CSO+RGO are shown in Appendix A.

Finally, the XPS spectrum of Co 2p present in the CSO of hybrid nanomaterials is shown in Appendix A. From both spectra, it is possible to observe that a narrow orbiting spin doublet (BE Co 2p_3/2_ = 780.1 eV; BE Co 2p_1/2_ = 796.2 eV) accompanied by a shake-up satellite on the main photoemission lines (786.2 eV and 802.2 eV) is present. The binding energy values and the appearance of strong satellite features are characteristic of the presence of Co^2+^ without the presence of other oxidation states, as was previously reported by us with similar oxides [3,11]. 

### 3.2. Electrochemical Characterization of Modified Glassy Carbon Electrodes with Hybrid Nanomaterial CSO/RGO and CSO+RGO

The behavior of the nanohybrid-modified electrodes and their individual materials was evaluated by cyclic voltammetry using ferrocene methanol as a redox mediator. The results show a clear synergistic effect between CSO and RGO in both *ex situ* and *in situ* materials (Appendix A). Greater currents compared with CSO alone and a shifting of the peak potentials towards lower energies compared to RGO, concomitant to a lower ∆Ep, indicate faster electron transfer kinetics with the hybrids materials.

Additionally, electrochemical impedance spectroscopy was used to characterize the modified electrodes. The experiments were performed in 10 mM hydrogen peroxide solution as a redox probe. Figure 3 shows Nyquist plots for RGO-, CSO-, CSO+RGO- and CSO/RGO-modified GCE electrodes. The data were plotted using the (R_s_(R_ct_CPES2)) equivalent circuit model (Inset: Figure 3), where R_s_, R_ct_ and CPE represent the resistance of the solution, the charge transfer resistance and the constant phase element, respectively. The Nyquist plot of hybrid nanomaterials exhibits much smaller semicircles than CSO and RGO. This indicates that the surfaces of the hybrid materials present lower interfacial resistance to the electron transfer process between the analyte and the electrode surface. Additionally, there is a synergistic effect between the material and the R_tc_ values. The charge transfer resistance (R_ct_) values are 6663 ± 750 Ω for RGO, 15,247 ± 861 Ω for CSO, 3491 ± 635 Ω for CSO+RGO *ex situ* and 2589 ± 184 Ω for CSO/RGO *in situ* modified GCE electrodes. *In situ* and *ex situ* hybrids presented similar resistance values between them. The capacitance values obtained from the CPE (Table 1), related to the active sites, were also similar between both the *ex situ* and *in situ* hybrid nanomaterials.

Our results are in agreement with those reported by Cao et al. who, studying diffusion processes using different carbon electrodes, observed that a greater resistance to charge transfer occurs in electrodes with less area and roughness and finally present cyclic voltammograms with less charge transfer speed (lower delta Ep) [36].

### 3.3. Amperometric H_2_O_2_ Detection

The electroanalytical response of the modified electrodes was evaluated by amperometry. Ten successive recordings were analyzed after the addition of 0.10 mM H_2_O_2_ in NaOH pH 12 solution. A detection potential of −0.400 V or 0.300 V for the reduction or oxidation of H_2_O_2_ was applied. Figure 4a shows the amperograms for the reduction of hydrogen peroxide using modified electrodes with the *in situ* and ex situ hybrids, CSO and RGO. The sensitivity to H_2_O_2_ detection normalized by the electroactive area was calculated from the amperograms (Figure 4b). A pronounced increase in the sensitivity with hybrid nanomaterials is observed with synergy by potentiation in their response since the values do not correspond to the sum of the sensitivities of CSO and RGO individually. The sensitivities of the CSO/RGO *in situ* and CSO+RGO *ex situ* nanohybrids are similar, in agreement with the impedance results.

From the calibration curves represented in Figure 4c, it was calculated that the sensitivity to analyte detection with the CSO/RGO in situ electrode was 74 ± 11 µA/mMcm^2^ and 80 ± 9 µA/mMcm^2^ with the CSO+RGO *ex situ* electrode. The detection limits (taken as 3.3 σ/S, where σ is the standard deviation of the blank signal and S is the sensitivity) were 7.7 µM and 7.4 µM for CSO+RGO *ex situ* and CSO/RGO *in situ*, respectively. 

Similarly, the response of the modified electrodes for the oxidation of hydrogen peroxide was investigated by applying a potential of 0.300 V. The current vs. time curves obtained are shown in Figure 4d after ten consecutive additions of 0.1 mM analyte. Under these conditions, an increase in current is observed after the addition of H_2_O_2_. In Figure 4e, the sensitivity to H_2_O_2_ detection normalized by the electroactive area is presented. No significant differences were observed between the sensitivities calculated for the electrodes modified with both hybrid nanomaterials. However, unlike the response of the analyte by reduction, in this case, there is no synergistic effect between the two nanomaterials.

From the calibration curves in Figure 4f, the sensitivity to the analyte detection was calculated, and values of 267 ± 15 µA/mMcm^2^ and 255 ± 13 µA/mMcm^2^ were obtained for the CSO/RGO electrode and CSO+RGO electrode, respectively. The detection limits calculated were 2.3 µM for both the CSO+RGO *ex situ* and CSO/*RGO in situ* electrodes, i.e., there were no differences between the two nanohybrids.

To evaluate the selectivity of the hybrid materials, amperometric recordings were taken by adding H_2_O_2_ aliquots followed by interfering species such as uric acid, glucose, ascorbic acid, and sodium sulfate at concentrations of 0.1 mM and 1.0 mM (Figure 5a,b). Both detection platforms showed great selectivity against the interfering species when we used the reduction mode of detection, with no detectable electrochemical signal. In contrast, when the oxidation mode is applied (Figure 5c,d), glucose and ascorbic acid present signals that can interfere in a future application to a real sample. Moreover, the reduction mode generates lower noise that can be useful in real sample applications.

Finally, the stability test performed over 1200 s (20 min) showed that the responses obtained using the *ex situ* hybrid material (CSO+RGO) have a stable signal for both the oxidation and reduction of hydrogen peroxide compared to the *in situ* hybrid material (CSO/RGO), where no stable signal was observed for oxidation with an increment of ~66% during the tested time (Appendix A).

Finally, comparing our systems with similar ones (Appendix A), it can be seen that for the detection of hydrogen peroxide by reduction, our electrodes are analytically similar and allow the use of the lowest potentials (−0.400 V) in a highly basic medium. Other systems reported in PBS pH 7.0 report either higher potential for detection or low sensitivities in relation to our systems.

## 4. Conclusions

Through hydrothermal synthesis, the nanostructured cobalt oxide CSO was generated, free of contaminants, alone and in the presence of RGO. Thus, two nanohybrids were tested for the electrocatalytic detection of hydrogen peroxide: the *in situ* nanohybrid, where CSO was synthesized in the presence of RGO, and the *ex situ* nanohybrid generated from the mechanical mixture between both materials.

In the case of CSO, there were no differences between both nanohybrids either by oxidation or by reduction, unlike what we previously observed with cobalt titanate CTO hybrids, in which the *in situ* nanohybrid clearly had the best performance for the electroreduction of hydrogen peroxide [3].

An explanation can be found in the fact that in the case of the CTO, the resistance to charge transfer value of the *in situ* hybrid was half that of the *ex situ* hybrid, whereas for CSO, this difference was only close to 30%.

Comparing the electrocatalysis of both CSO hybrids, there was clearly synergy when the reduction mode was used and only a sum of effects in the case of oxidation. In addition, the study of interferents did not show an effect in the reduction mode but it did in the oxidation mode. All these facts combined with the low experimental noise present in the first mode and more stable signals allow us to conclude that the detection of hydrogen peroxide would be more efficient via the electroreduction of the analyte.

## Figures and Tables

**Figure 1 micromachines-14-01059-f001:**
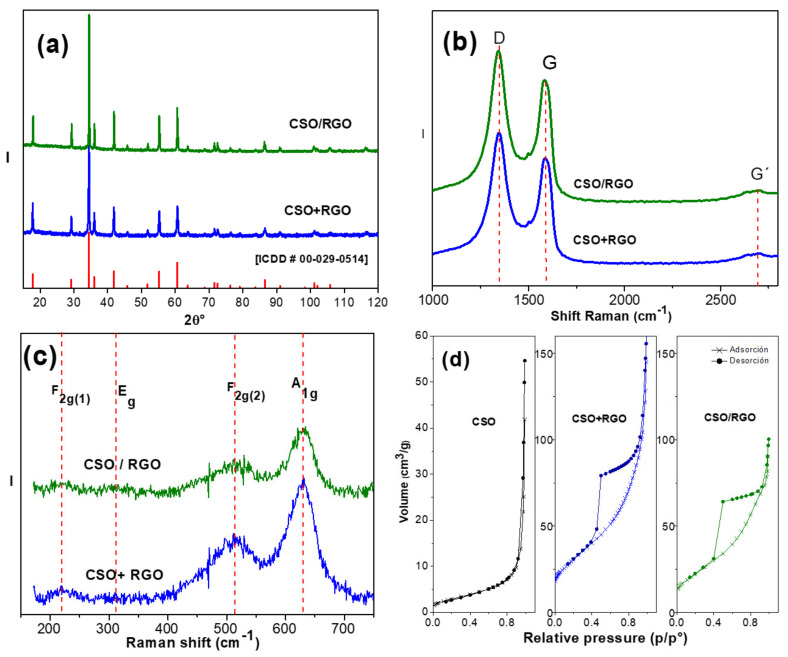
(**a**) Phase analysis of XRD patterns [ICDD #00-029-0514]. (**b**) Raman spectra between 1000 and 3000 cm^−1^ Raman shift. (**c**) Raman spectra between 200 and 750 cm^−1^ Raman shift. (**d**) N_2_ adsorption–desorption isotherm profile.

**Figure 2 micromachines-14-01059-f002:**
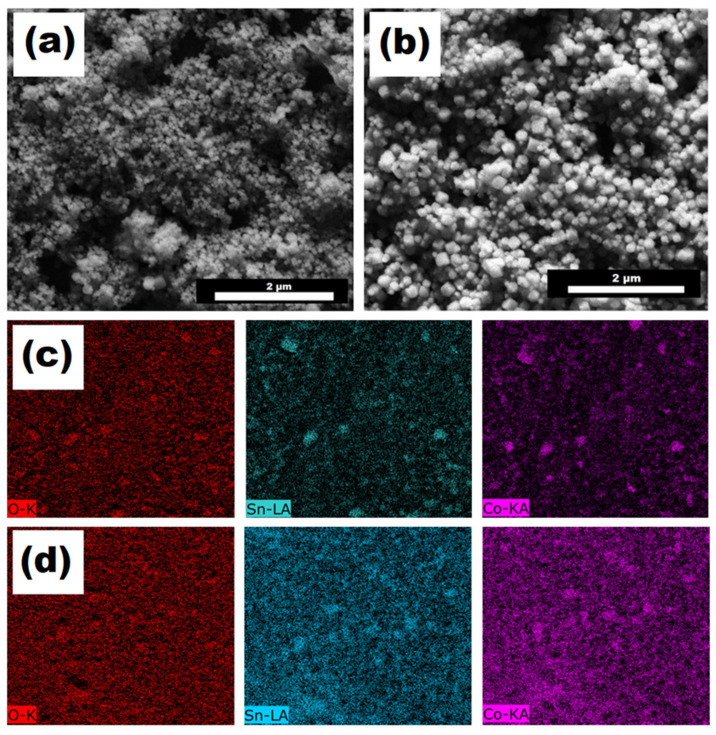
SEM micrographs of (**a**) CSO+RGO and (**b**) CSO/RGO. EDX mapping (oxygen, tin, and cobalt) of (**c**) CSO+RGO and (**d**) CSO/RGO.

**Figure 3 micromachines-14-01059-f003:**
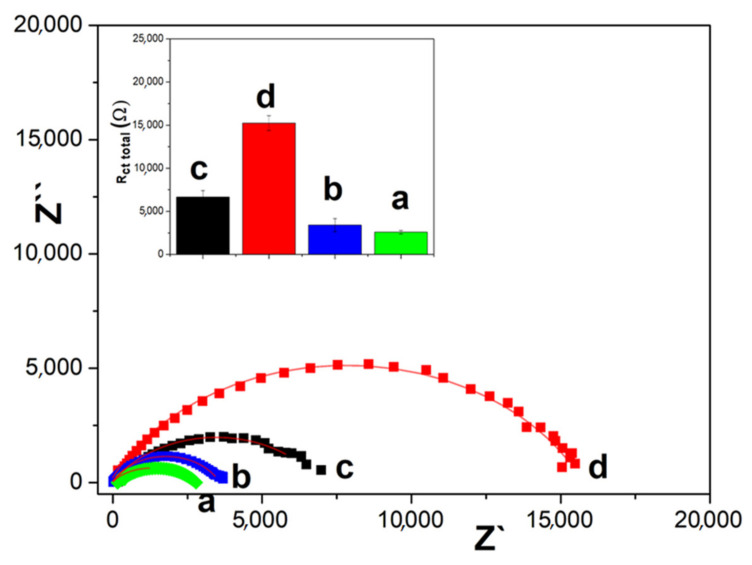
Nyquist plot of 10 mM hydrogen peroxide solution as a redox probe on: a. CSO/RGO, b. CSO+RGO, c. RGO, d. CSO. Figure Inset: Total resistance and equivalent circuit.

**Figure 4 micromachines-14-01059-f004:**
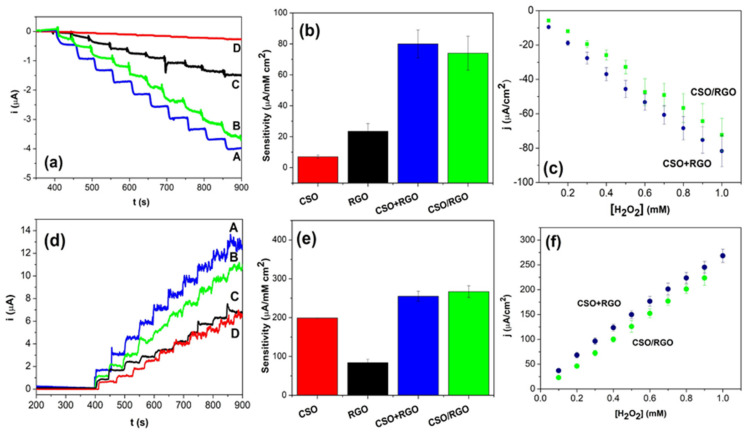
(**a**) Amperometric response of A. CSO+RGO, B. CSO/RGO, C. RGO and D. CSO at −0.400 V; (**b**) sensitivity toward H_2_O_2_ reduction obtained from amperometric experiments; (**c**) calibration plot for the electrochemical H_2_O_2_ reduction response. (**d**) Amperometric response of A. CSO+RGO, B. CSO/RGO, C. RGO, and D. CSO at 0.300 V. (**e**) Sensitivity toward H_2_O_2_ oxidation obtained from amperometric experiments. (**f**) Calibration plot for the electrochemical H_2_O_2_ oxidation response. Amperometric experiments were performed with the successive addition of 0.1 mM H_2_O_2_ in NaOH solutions at pH 12.

**Figure 5 micromachines-14-01059-f005:**
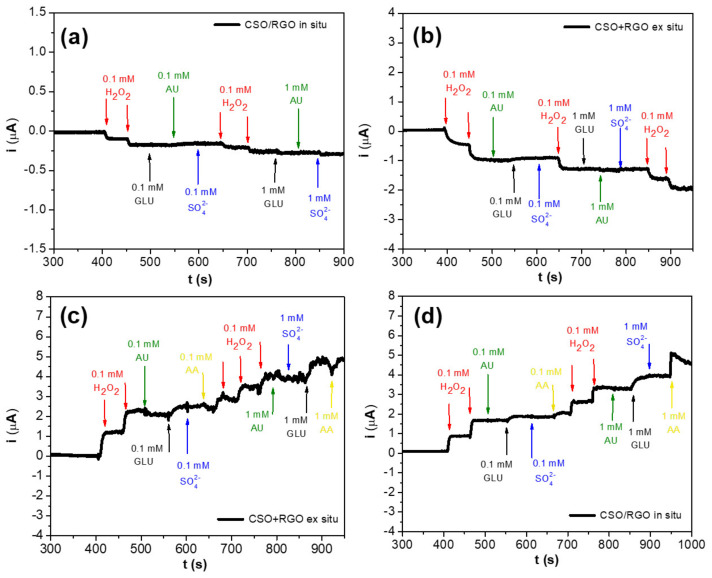
Amperometric responses to successive additions of 0.1 mM and 1.0 mM H_2_O_2_, uric acid (AU), glucose (GLU), sodium sulfate (SO_4_^2−^) and ascorbic acid (AA): (**a**) reduction for CSO/RGO *in situ* and (**b**) reduction for CSO+RGO *ex situ* electrodes at −0.400 V applied, (**c**) oxidation for CSO+RGO *ex situ* (**d**) oxidation for CSO/RGO *in situ* electrodes at 0.300 V applied.

**Table 1 micromachines-14-01059-t001:** Crystallite size, textural analysis, and EIS parameters for CSO+RGO and CSO/RGO.

	Crystallite Size (nm)	Surface BET (m^2^/g)	Total Pore Volume (cm^3^/g)	Pore Size (nm)	C (10^−5^ F)	R_tc_ (Ω)	k_ET_ (10^−4^ cm s^−1^)
CSO+RGO	29.5 ± 4.8	109	0.243	47 ± 16	2.85	3491 ± 635	2.46
CSO/RGO	38.9 ± 7.8	82	0.152	31 ± 11	3.09	2589 ± 184	2.11

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
