# Peer review of "In situ* or *Ex situ* Synthesis for Electrochemical Detection of Hydrogen Peroxide—An Evaluation of Co_2_SnO_4_/RGO Nanohybrids"

_micromachines, 2023, doi:10.3390/mi14051059_

Round 1
Reviewer 1 Report
In this manuscript, Co2SnO4/RGO(Reduced Graphene Oxide) nanohybrids were prepared using two-type methods. Readers will be interested in the differences in physical properties and electroanalytical response between the two preparation methods.
However, there are some questions to be solved as follows.
・The notations of (a) CSO+EGR and (b) CSO+RGO in the captions of Fig. 2 are the same.
・Are c and d in Fig. 3 reversed?
・Did the pore volumes and pore sizes affect the electroanalytical response (ex., diffusion)?
Author Response
Many thanks to the reviewer, his comments were a great contribution to improve the work.

Reviewer 2 Report
After carefully reading the revised version of this manuscript, I found the research therein to be of some use. Submitted works may be published with significant revisions.
1 The authors should provide XPS analysis profiles of various materials.
2 The SEM image in Figure 2 should be clearer
3 The curve of the glassy carbon electrode alone is not available in Figure 3, meanwhile, the resistance of RGO seems to be a bit too high, is the reduction of GO by the authors successful?
4 The authors should provide the stability and reproducibility of the sensor. It is also not indicated whether the surface of the modified electrodes changes, e.g. corrosion, loss of surface material, etc.
5 The authors did not compare the sensors with those of other workers.
6 TheAuthors should improve the English writing in their manuscripts
After carefully reading the revised version of this manuscript, I found the research therein to be of some use. Submitted works may be published with significant revisions.
1 The authors should provide XPS analysis profiles of various materials.
2 The SEM image in Figure 2 should be clearer
3 The curve of the glassy carbon electrode alone is not available in Figure 3, meanwhile, the resistance of RGO seems to be a bit too high, is the reduction of GO by the authors successful?
4 The authors should provide the stability and reproducibility of the sensor. It is also not indicated whether the surface of the modified electrodes changes, e.g. corrosion, loss of surface material, etc.
5 The authors did not compare the sensors with those of other workers.
6 TheAuthors should improve the English writing in their manuscripts
Author Response

(The authors gave the same response as above.)

Reviewer 3 Report
This paper compares the properties of the Ex situ synthesis and In situ synthesis of Co2SnO4/RGO, where Ex situ synthesis is mechanical mix of the two components, and In situ synthesis is to mix in the solution. The authors compared the characterizations by XRD, SEM, Raman, and isotherm, and the electrochemical properties by EIS and Amperometric detection of H2O2.
Below are my comments:
1. Overall from the manuscript, the electrode material is used for detecting H2O2, so I suggest the title be modified to be more specific to H2O2
2. The SEM figure, both were label as CSO+RGO (I assume b is CSO/RGO). From the SEM I can see a rough surface of carbon on a micrometer scale, and CSO/RGO is rougher. Recent research has focused on the geometric effect of electrode materials. Is this one of the reasons CSO/RGO performs better?
some references:
Cao, Q., Puthongkham, P., & Venton, B. J. (2019). new insights into optimizing chemical and 3D surface structures of carbon electrodes for neurotransmitter detection. Analytical Methods, 11(3), 247-261.
Cao, Q., Shao, Z., Hensley, D. K., Lavrik, N. V., & Venton, B. J. (2021). Influence of geometry on thin layer and diffusion processes at carbon electrodes. Langmuir, 37(8), 2667-2676.
Kousar, A., Peltola, E., & Laurila, T. (2021). Nanostructured geometries strongly affect fouling of carbon electrodes. ACS omega, 6(40), 26391-26403.
Pascual, L. F., Pande, I., Kousar, A., Rantataro, S., & Laurila, T. (2022). Nanoscale engineering to control mass transfer on carbon-based electrodes. Electrochemistry Communications, 140, 107328.
Krivić, D., Vladislavić, N., Buljac, M., Rončević, I. Š., & Buzuk, M. (2022). An insight into the thin-layer diffusion phenomena within a porous electrode: Gallic acid at a single-walled carbon nanotubes-modified electrode. Journal of Electroanalytical Chemistry, 907, 116008.
3. The electrochemical characterization has only EIS, which is relatively weak. I would suggest the authors add CV in different analytes (ferricyanide, Ru(NH3)6, etc.) and other relative characterizations.
4. The EIS plot inset label is not consistent with the figure. Can the authors clarify which solution did the author use to test EIS? Also, charge transfer is usually written to be Rct.
5. The sensor needs a stability test.
Minor editing of English language required
Author Response

(The authors gave the same response as above.)

Round 2
Reviewer 2 Report
accept